# Approach to standardized material characterization of the human lumbopelvic system—Specification, preparation and storage

**Sascha Kurz** [1‡]*, **Marc Gebhardt** [2,3‡], **Fanny Grundmann** [4], **Christoph-Eckhard Heyde** [1], **Hanno Steinke** [3]

**1** ZESBO - Center for Research on Musculoskeletal Systems, Department of Orthopedic Surgery, Traumatology and Plastic Surgery, Faculty of Medicine, Leipzig University, Leipzig, Germany, **2** Institute of Experimental Mechanics, Faculty of Civil Engineering, Leipzig University of Applied Sciences, Leipzig, Germany, **3** Institute of Anatomy, Faculty of Medicine, Leipzig University, Leipzig, Germany, **4** Clinic of Trauma, Orthopedic and Septic Surgery, Hospital St. Georg gGmbH, Leipzig, Germany

‡ SK and MG are contributed equally to this work as co-first authors.
* sascha.kurz@medizin.uni-leipzig.de

**Data Availability Statement:** The data used for this article are included in the Supporting information.

## Abstract

The complexity of the osseo-ligamentous lumbopelvic system has made it difficult to perform both, the overall preparation as well as specimen harvesting and material testing with a reasonable amount of time and personnel. The logistics of such studies present a hurdle for reproducibility. A structured procedure was developed and proved, which allows all necessary steps to be carried out reproducibly and in a reasonable time. This enables the extraction of 26 soft tissue, 33 trabecular and 32 cortical bone specimens from this anatomical region per cadaver. The integrity of the specimens remains maintained while keeping requirements within manageable limits. The practicability of the intended five-day specimen harvesting and testing procedure could be demonstrated on five test and two pre-test sequences. The intended minimization of physical, biological, and chemical external influences on specimens could be achieved. All protocols, instructions and models of preparation and storage devices are included in the supporting information. The high grade of applicability and reproducibility will lead to better comparability between different biomechanical investigations. This procedure proven on the human pelvis is transferable to other anatomical regions.

## Introduction

Due to its high geometric complexity and clinical relevance, the human pelvis is an ideal example for the development of a standardized preparation procedure for mechanical material specimens. The lumbopelvic system fulfills a central role in load transfer and redirection between the torso and lower extremities [1–3]. The ligamentous tightened ring structure consisting of three primary bone segments sometimes challenges surgeons during fracture treatment. Due to the relevance for locomotion, the quality of life is also dependent on a maintained biomechanical function of the pelvis [4, 5].

Body donor-related data were anonymized to prevent inferences about their identity.

**Funding:** This study was funded by the Federal Ministry for Economic Affairs and Energy (grant numbers MG: ZIM 16KN051655, SK: ZIM 16KN051656), the Saxon State Government out of the state budget approved by the Saxon State Parliament (stipend reference MG: 31004 70 809) and Open Access Publishing Fund of Leipzig University. The funders had no role in study design, data collection and analysis, decision to publish, or preparation of the manuscript.

**Competing interests:** The authors have declared that no competing interests exist.

To better understand the complex processes in the context of osseous and ligamentous load transfer, numerical simulations are used in anatomy and medicine to analyze the influence of the osseo-ligamentous system on fracture mechanics and for fracture mechanics planning complex reconstructions and corrections of severe pelvic trauma [4]. Here, in particular, the finite element method can be of great support for the qualitative assessment of different osteo-synthesis variants [6–8]. Essential for the quality of these simulations, is the knowledge of the underlying material properties of all involved structures. Dalstra and Huiskes [9] and later Lin-strom's group [10] already indicated clear tendencies of load dissipation within the pelvis. Based on Wolff's law [11] it can therefore be deduced, that these primarily load-dissipating regions also provide structural peculiarities compared to the less loaded regions, in order to be able to achieve the necessary physiological and therefore anatomical function. Thus, these areas identified as particularly relevant should be characterized in more detail. In addition to anatomically and mechanically important regions, this also applies to surgical areas, e.g. anchor-stable positions. In order to perform qualitatively appropriate numerical analyses, not only the knowledge of the particular material parameters of the simulated structures are necessary, but also the knowledge of the area-wise parameter variation within this structure. Due to the complexity described above, such investigations are very rare and, if available, difficult to compare or reproduce. A sensitivity analysis was provided by Anderson, Peterson et al. [12] which showed that, when comparing experimental and simulation results, the greatest influence lies in the thickness and elasticity of the cortical bone. The aforementioned studies could not be based on pelvis-specific material parameters because they are scarce. A broad study on region-specific mechanical properties of the pelvis was done by Dalstra and Huiskes [13], which focused on the elastic properties of the trabecular bone. In comparison, the procedure presented here also addresses cortical bone and soft tissue. Furthermore, the individual position and local direction (topology) of each specimen are unambiguously defined. Each individual topology is completely transformable into any super- and subordinate reference systems. This allows a topologically (position, direction, geometry) defined harvesting and allocation of the material properties.

In addition, if the development of a numerical model based on the preparation is envisaged, a prior non-destructive investigation of the entire preparation is recommended. The combination of the geometry, the obtained material data, as well as a deformation analysis, such as Range of Motion [14], enables to train, calibrate, or validate the model. However, since it is difficult to carry out such complex investigations, the structuring and logistics represent a crucial hurdle to reproducibility. This includes experiments on the entire preparation as well as the harvesting of samples and the performance of material experiments. In particular, this is difficult within a reasonable period of time, with feasible personnel and high demand for quality. In addition, all this is considered to be amplified by the neither inert nor time-stable character of the investigated material. This methodological problem is addressed here.

## Methods

### Preparation acquisition

The methodology described in this section was successively developed and tested on seven adult human cadaver pelvises. The pelvises examined included the lumbopelvic system up to the third lumbar vertebra with a mostly intact connective tissue apparatus. All cadavers originated from the Institute of Anatomy of Leipzig University. During their lifetime, all of them had given written consent to dedicate their bodies to medical education and research purposes. Being part of the body donor program regulated by the Saxonian Death and Funeral Act of 1994 (3rd section, paragraph 18, item 8), institutional approval for the use of the post-mortem

tissues was obtained from the Institute of Anatomy, Leipzig University. The authors declare that all experiments were performed according to the ethical principles of the Declaration of Helsinki.

## Condition

In accordance with the recommendations of Martin and Sharkey [15], avoidable influences on the mechanical properties of the specimens were prevented when possible. Their observations indicate a clear influence of chemical treatment. Conversely, they do not ascribe any significant effect to freeze-thaw cycles.

Furthermore, in a previous study, the influence of the material sample condition on the determination of the material properties could be demonstrated [16]. As expected a priori, the best possible results were always obtained with fresh and unfixed specimens. Fresh-frozen specimens showed only minimal changes in mechanical properties.

However, since general delays in both cadaver acquisition and test preparation can be expected, the fresh-frozen condition is of high relevance in biomechanical research. According to studies by Lee, Oswald, Jung, and others [17–21], the freeze/thaw cycle, required to work with fresh-frozen preparations, has no noticeable effect on the biomechanical properties of soft tissues. Unger et al. [22] showed, that there is no relevant influence on osseous structures. As the use of fresh-frozen preparations can therefore be regarded as legitimate for the intended investigations, such specimens were used for the subsequent studies. Preservative anatomical fixation, for example in the form of ethanol or formalin fixation, was also completely avoided. Although this would make the tissue more durable and kill any pathogens, Hammer et al. [23] demonstrated, that the associated changes to the organic matrix would result in irreversible changes to the material properties.

## Procedure design

In order to be able to manage the problem of structured logistical test planning, described at the beginning, whereby non-destructive tests on the entire preparation, as well as material sample extraction and testing of these, are planned, a structured schedule was developed. This allows all necessary steps to be conducted within a time span that should be insignificant for the integrity of the specimen, while at the same time keeping the personnel and the equipment requirements manageable. With this procedure design, it is possible to run all mentioned steps with two to three persons within five working days (usually 8 hours, extendable to 10 hours). In addition, only one set of equipment, for example, the testing device is required, as adequate time is available for the test setup conversion. The optional break, enabled by shock freezing the osseous segments before harvesting osseous specimens, allows even more freedom in the design of experiments, without causing an unpredictable influence on the material samples. Parallelizing the procedure would allow further acceleration, yet this reduces the applicability due to higher requirements. The procedure scheme is shown with the corresponding temperature curve in Fig 1. The procedural points mentioned herein are explained in chronological detail in the subsequent sections. All afterward mentioned steps are defined in a standard operating procedure (SOP). The entire SOP (S1 File), including appendices as well as all 3D models of auxiliaries (S2 File), are attached as supporting information.

## Pretreatment

Immediately after dissection at Leipzig Anatomy, the cadaver preparations were wrapped in polyethylene foil to prevent them from drying out and cooled down to approximately 4 ˚C. After the size-dependent cooling duration, the preparations were immediately transferred to

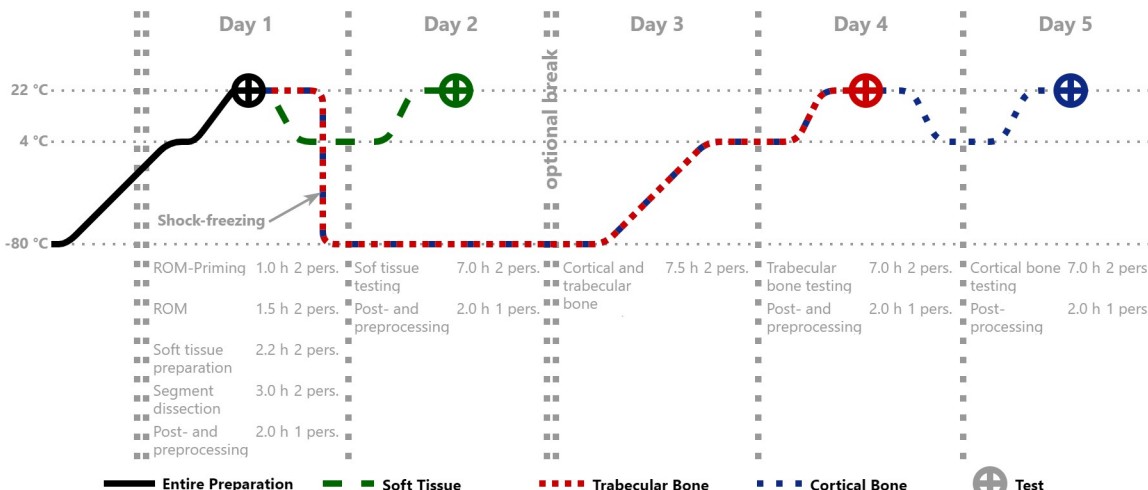

**Fig 1. Procedure design.** Procedure design flow chart with corresponding temperature curve of preparations as well as estimated time and personnel (Day 1: testing of entire preparation, harvesting of soft tissue specimens, segment dissection for bone specimens; Day 2: testing of soft tissue specimens; Day 3: harvesting of osseous specimens; Day 4: testing of trabecular bone specimens; Day 5: testing of cortical bone specimens; left open side: deep-frozen storage period).

the approx. -80 ˚C cooling and embedded in -80 ˚C cold packs, which were changed after five minutes. The storage period until testing was also carried out at approx. -80 ˚C. To minimize the time and effort required, all preparations were subjected to a patient computed tomography (CT) scan at approx. -80 ˚C during the same day. This allowed their individual osseous geometries to be determined and their cortical and trabecular layer thicknesses estimated. Low influence tests, which can be performed on the entire preparation in the deep-frozen state, should be performed during the storage period (Fig 1, left open side) while avoiding interruption of the cold chain.

72 hours before the testing started, the thawing process was initiated by transferring the cadaver preparations to a 4 ˚C refrigerator (Fig 1). Starting with the embedding of the pelvis at the third lumbar vertebra to perform an optional non-destructive Range of Motion analysis, the examinations of the entire preparation were initiated. Tests such as these, for example for verification purposes, can be considered optional for the procedure described here. To exclude falsifying influences, no specimens should be harvested from the embedding area. This applies in particular to embedding processes in which tissue alteration cannot be precluded, for example, due to exothermic reactions during embedding.

## Soft tissue preparation

As the integrity of the ligamentous apparatus had to be ensured during dissection, the preparations contained plenty of excess soft tissue, mostly muscles. The removal of this excess biological material required knowledge of the exact anatomical location of the targeted specimens to ensure their integrity. For this purpose, tissue scissors and a scalpel were used. In the vicinity of the specimens, a so-called blunt preparation is recommendable. Here, the tissue remnant to be removed is carefully scraped off using the side of the scalpel facing away from the blade. Subsequently, the relevant and testable ligamentous and fascial structures were dissected. An exemplary soft tissue harvesting is shown in Fig 2. The detailed chronological harvesting procedure and the recommended specimen designations can be found in the SOP (S1 File). The collected soft tissue specimens were immediately moistened with a 0.9% saline solution,

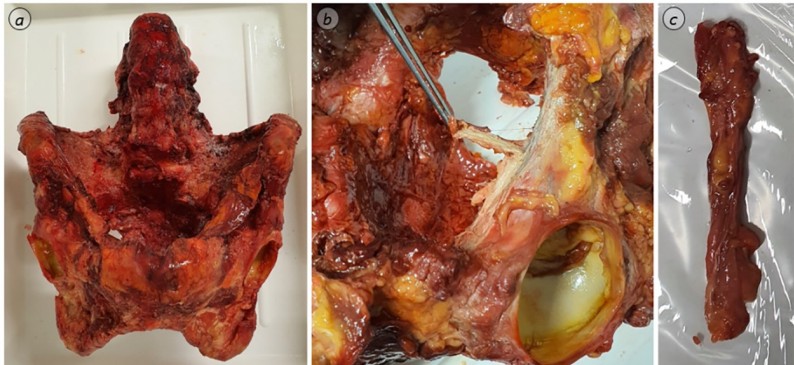

**Fig 2. Soft tissue dissection.** Exemplary soft tissue specimen dissection process (*a*: pelvis including ligamentous apparatus; *b*: removal of the Ligamentum pectineum; *c*: Ligamentum pectineum from cranioventral).

wrapped in polyethylene film, and labeled. The specimens prepared in this way were gathered directly in an airtight container and intermediately cooled. Thus, all soft tissue specimens were stored refrigerated at 4 °C until testing on the following day (Day 2, Fig 1).

## Segment dissection

After the removal of all remaining soft tissue components (compare Fig 3e and 3f), the osseous segments were dissected. These were predefined based on anatomically defined as well as surgically relevant areas. Furthermore, special requirements for the specimen geometries as well as repeatable sectioning had to be ensured. Their compliance was verified by analysis of all cadaver CT data. In order to classify these sections relevant for extraction, easily palpable and locatable anatomical landmarks were defined. Planes were drawn according to these landmarks, along which the osseous lumbopelvic system was sectioned using an oscillating saw.

The sectioning planes and the resulting segments are described in detail in the visual harvesting protocol embedded in the SOP (S1 File) as well as in Fig 3. An overview of this segmentation is shown in Fig 3a and 3b, and a literal description of the sectioning planes can be found in Table 1. For example, the plane labeled as 5, which separates the Ala ossis ilii inferior (labeled as r2) and Corpus ossis ischii (labeled as r3), is determined normal to the cortical surface by Spina iliaca anterior inferior and Incisura ischiadica major. These segments were then shock frozen to approx. -80 °C with a 75% / 25% acetone-water-coolant solution following the procedure described by Steinke [24]. Where required, this allows the experimental procedure to be split and thus more variable planning; it also prevents denaturation of the tissues and allows improved downstream processing.

An additional test was performed on a human lumbar vertebra to examine the low influence of the presented shock freezing procedure. For this purpose, a comparative weighing (ML303T/00, Mettler-Toledo, Greifensee, Switzerland) of the samples, as well as the pycnometric analysis (Pycnometer, borosilicate glass, B 50 ml) of coolant solution samples were performed. In addition, an optical evaluation of these solution samples and the residues was performed after an evaporation test. The density of the coolant increased only by 1.32% after 1.5 hours of immersion of a lumbar vertebra. The lumbar vertebra did not undergo any relevant change in mass (0.11%). Afterwards, the vertebrae were cut into thin slices and shock frozen again. Analogous to the whole vertebrae, the test on the open-pored trabecular slices also showed no relevant changes (density change of coolant 0.92%, mean mass change of slices

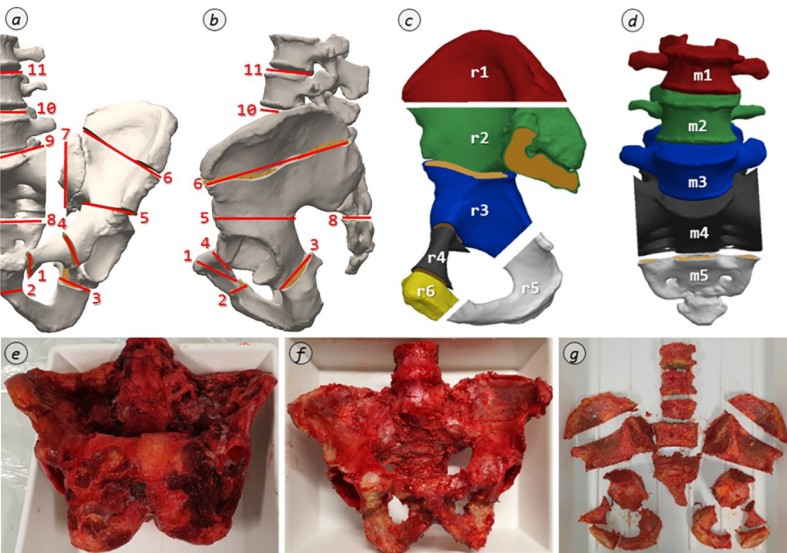

**Fig 3. Segment dissection.** Exemplary segmentation based on the visual harvesting protocol (*a*: definition of the sectioning planes, ventral view; *b*: definition of the sectioning planes, lateral view; *c*: segments definition of the right ilium region r; *d*: segment definition of the middle region m; *e*: pelvis including ligamentous apparatus; *f*: pelvis freed from soft tissue; *g*: segmented pelvis).

0.21%). This indicates that no relevant dissolution phenomena of the examined bone tissue segments in the coolant solution are to be expected during the shock freezing procedure.

## Harvesting of osseous specimens

Within the developed visual harvesting protocol (S1 File), all specimen positions as well as their local coordinate systems and specific designations are listed. Fig 4 illustrates dissecting the second segment of the left pelvic half as an example. The specimen geometry and

**Table 1. Detailed description of sectioning planes.**

| No. | Incision | Definition |
|---|---|---|
| 1 | Lower parasymphyseal | Tuberculum pubicum, parallel to Symphysis pubica |
| 2 | Ramus ossis ischia | Ramus inferior ossis pubis to Symphysis pubica, parallel to the inferior end of the Symphysis pubica |
| 3 | Upper parasymphyseal | Foramen obturatum through Corpus ossis ischii, parallel to the inferior end of the Acetabulum, superior to the Spina ischiadica |
| 4 | Ramus superior ossis pubis | Corpus ossis pubis, from Tuberculum between Corpus and Ramus superior ossis pubis, normal to Ramus superior ossis pubis |
| 5 | Supraacetabular | Spina iliaca anterior inferior to Incisura ischiadica major, through Sulcus supraacetabularis |
| 6 | Ala ossis ilium | Spina iliaca anterior superior to Spina iliaca posterior superior, normal to Fossa iliaca |
| 7 | Os sacrum, Pars lateralis | Between Crista sacralis lateralis and Articulatio sacroiliaca, parallel to Sagittal plane |
| 8 | Os sacrum | Lineae transversae between Corpus vertebrae sacralis 2 and 3 |
| 9 | Basis ossis sacri | Separating Corpus vertebrae 5 from Os sacrum, through Discus intervertebralis |
| 10 | Intervertebralis 4/5 | Separating Corpus vertebrae 4 and 5, through Discus intervertebralis |
| 11 | Intervertebralis 3/4 | Separating Corpus vertebrae 3 and 4, through Discus intervertebralis |

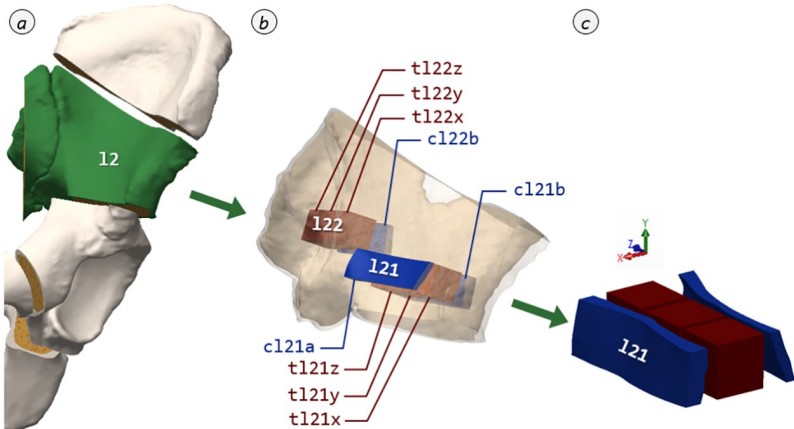

**Fig 4. Bone specimen dissection.** Partial segmentation according to visual harvesting protocol using region l2 as an example (*a*: segmented pelvis with marked segment l2; *b*: segment l2 with harvesting locations l21 and l22; *c*: detailed view of harvesting location l21).

dimensions were determined based on the characteristics of the human pelvis, but also on the requirements of the material testing. These are geometric factors, such as the curvature of the pelvic wings and the area diameter of the trabecular interlayer, as well as the minimum dimensions necessary to perform material testing. As a result, cortical beams with their natural thickness, an aspired width of 10 mm and a length of about 40 mm proved to be suitable. For trabecular bone cubes an edge length of about 10 mm was appropriate. Likewise, this should satisfy the continuum assumption of Harrigan et al. [25], where five intertrabecular distances must be contained in each dimension. To ensure the feasibility of specimen harvesting even with smaller pelvis geometries, Computer-Aided Design software (SolidWorks 2015, Dassault Systèmes, Vélizy-Villacoublay, France) was used. Based on two vastly different reference models of the pelvis in terms of shape and size, the locations of the harvesting points within the segments were thereby visualized and defined. Thus, a positioning could be determined at which reliable specimen harvesting could be ensured.

Subsequently, preliminary tests were conducted to verify the practicability of harvesting at this location and to ensure that the specimens taken in this manner meet the defined specifications for material testing. To obtain the osseous specimens for material testing, the deep-frozen segments were cut individually using a diamond thin-cutting saw (Type 36, EXAKT Advanced Technologies GmbH, Norderstedt, Germany). To provide simplified guidance, an adjustable fence was specifically developed and manufactured using Fused Deposition Modeling (FDM), allowing a free angle adjustment between ± 45° and a 10 mm interval offset; specific FDM parameters are included in the S2 File. Using this, a straightening cut was first made in each case and the segment was then divided into the 10 mm wide harvesting areas. Subsequently, the peripheral regions were cut off and the remaining bars were cut into cubes for the harvesting of the trabecular bone in a sawing jig, which was also produced with FDM. By processing in a frozen state and not using rinsing water during the sawing process, a change in the bone marrow content was prevented.

During the entire cutting procedure, as exemplarily shown in Fig 5, the initial orientation must be tracked for each tilt and rotation to ensure the allocation of the resulting specimens. For the exemplary dissection sequence shown in Fig 5, this means: In the initial situation (Fig 5a), the local x-axis (in this case ventral) points forward-down, while the y-axis (cranial) points up and the z-axis (lateral right) to the left. By tilting the vertebra to remove the endplate

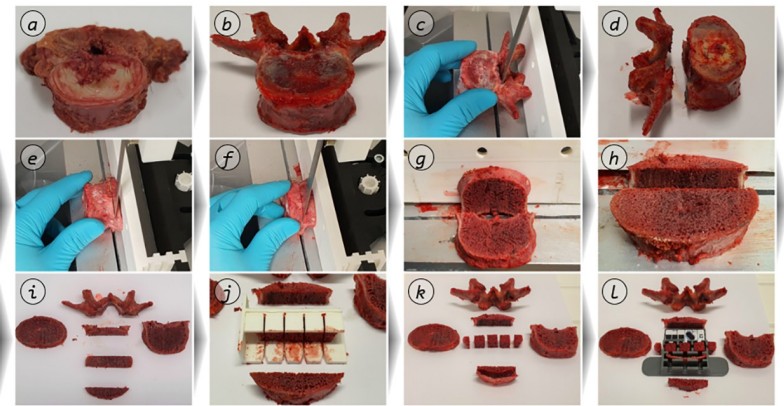

**Fig 5. Exemplary dissection sequence.** Dissection of a lumbar vertebra with auxiliaries (*a*: coarsely prepared vertebra; *b*: vertebra freed from soft tissue; *c*: segments that interfered with handling are removed in one cut; *d*: separated pure vertebral body; *e*: cranial opening cut of the trabecular bone, defining the plane normal to the local y-axis; *f*: cut offset by adjusting the saw fence by one step = 10 mm and creation of the harvesting plane; *g*: harvesting plane as disc; *h*: additional opening cut by adjusting the saw fence to define the plane normal to the x-axis; *i*: trabecular bone bar after additional cut and previously separated parts; *j*: trabecular bone bar inserted into the cutting guide; *k*: trabecular cube sawed by cutting along the local z-axis and ventrally separated cortical bone freed from the remaining trabecular bone; *l*: specimens placed in the prepared storage box).

(Fig 5e), the local x-axis points up and the y-axis points right. When completely dissected (Fig 5k), the local axes correspond to the initial situation again (Fig 5a). The cortical beams contained in the removed peripheral regions were then freed from remaining soft tissue remnants by scalpel and the remaining trabecular bone was disposed via Stille-Ruskin Bone Rongeur. In a final step, these were cut to the necessary specimen length according to the visual harvesting protocol. For storage of the material specimens, special storage boxes were developed (Figs 5l and 6), which clearly identify the specimen designation as well as the local harvesting axes. The boxes are equipped with a water reservoir in their base to create a water-saturated environment without direct drenching. This preserves the moisture state of the specimens.

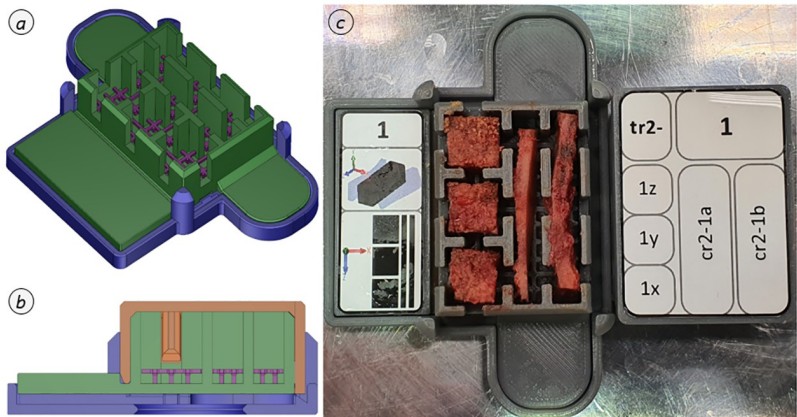

**Fig 6. Osseous specimen storage.** Custom-built storage box, combined type (*a*: construction drawing in ISO view; *b*: sectional view; *c*: osseous specimens in designated storing areas).

## Quality control

For quality surveillance assessment codes were defined, which include all necessary error points through the procedure, as well as their strengths. An assessment code combines keys for procedure step, assessment type and strength. For example, severe damage during sampling would result in the code "B02.3". Where "B" corresponds to the procedure step "Dissection", "02" to the assessment type "Damaged" and "3" to the strength "Complete". This enables the quantification and tracking of sample failures or deviations. The assessment code scheme is attached as an annex to the S1 File.

To ensure the accuracy of the preparation procedure, the shape quality of the harvested specimens was controlled. Also, the deviation of the obtained from the perfect shape for the material test was compared. This is most relevant for the trabecular bone specimens. Therefore, the actual cuboid volume of the trabecular bone specimens is compared to the imaginary perfect cube volume, given by the average edge lengths of each. The resulting value describes the percentage deviation from a perfect cube shape. This value is hereinafter defined as cubicity.

Since the thickness of the cortical and soft tissue specimens depends on their natural shape and shall be maintained, only the requirements for the mechanical testing devices are relevant. A minimum length and, for cortical and fascial specimens, a target width is related to this. In the test devices utilized here, these are 10 mm in width. The natural cross-section of the ligaments is to be preserved.

# Results

## Procedure assessment

A methodology for the lumbopelvic system was developed and approved. This allows a topologically defined harvesting (position, direction, geometry) for the first time. Through systematic dissection and designation, each specimen's topology and material type is completely specified. Every individual topology is completely transformable into any super- and subordinate reference systems. This procedure is optimized to minimize chemical, physical and biological influences on the material properties. Concurrently, it is as well optimized to enable the implementation with a justifiable expenditure of time, personnel and process resources. In particular, it allows the work steps of each day within the developed five-day procedure to be carried out by two people during a typical eight-hour day in a safe, reproducible and traceable manner. Auxiliaries have been developed for this purpose and are provided within the S2 File. Particular effort was given to design the procedure reproducible and transferable to other anatomical regions as well.

By applying the presented procedure to five test subjects, it was possible to successfully extract 88 of 130 possible soft tissue specimens, 146 of 165 trabecular, and 145 of 160 cortical bone specimens. Overall main reasons for absence are: dissected due to prior preparation, not dissected because of strong anatomical abnormality and failure through cutting plane error. For the osseous specimens, the most frequent error was an anatomical abnormality and geometrical imperfections. Anatomical abnormality means, for example, there were ossified or cancerous regions, or large voids, inside the trabecular bone matrix. The assessment code combinations, which lead to exclusion for all investigated tissues over all test subjects, are shown in Fig 7.

## Geometrical quality control

The aforementioned cubicity of the trabecular bone specimens was determined to be 99.03% on average with a standard deviation of 1.22% (range: 93.22–100.00%). The mean width was 9.55 mm ± 0.99 mm. For the cortical bone specimens, the only slight relevant geometric

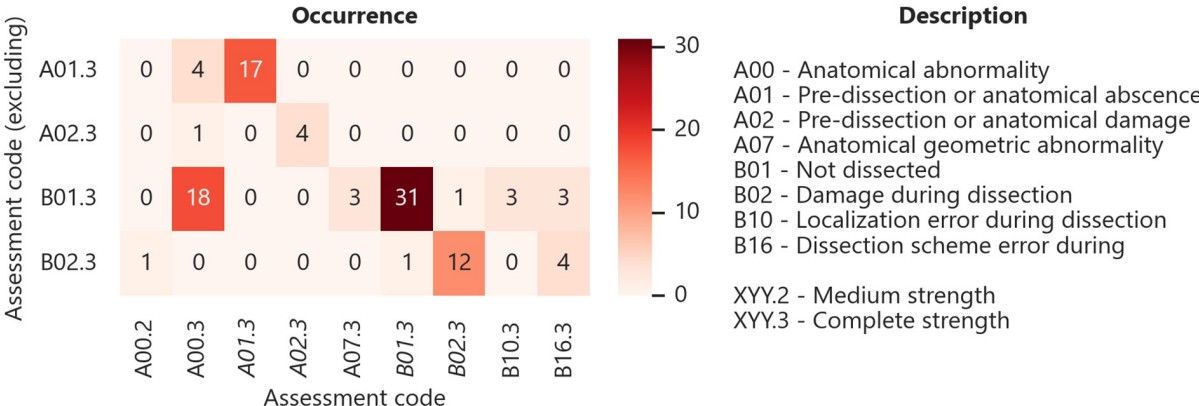

**Fig 7. Evaluation of excluding assessment code combinations.** Occurrence of assessment code combinations, which lead to exclusion, for all tissues.

dimension is the width, which results in 9.49 mm ± 1.17 mm. For the soft tissue samples, only the width of the fascial specimens is marginally relevant, which was 11.46 mm ± 2.81 mm. This was examined because a target ratio of width to test length (20 mm provided) of 0.5 is desirable for mechanical axial tensile tests. Compliance with the minimum length was found to be without difficulty and does not require any further consideration. For a more detailed illustration of the geometrical quality controls mentioned above, their distributions are shown in Fig 8. As clearly visible in the upper right corner, the main deviation of the trabecular bone specimens' edge length occurred in their individual local y-direction. This corresponds to their expansion normal to the cortical layer and depends on their natural thickness. The distribution of the cortical bone specimens' width along the intended span indicates the achievement of good parallelism.

As no mechanical post-processing is used for the reasons mentioned, geometrical deviations must be included in the underlying algorithm of a subsequent test evaluation. Fig 9 illustrates the occurring thickness distribution, as well as the categorization according to cadaver and harvesting regions. A subsequent Kruskal-Wallis H-test significantly indicates that these do not belong to a common population with respect to the region ($H_{6,138} = 36.082$, $p = 2.657 \cdot 10^{-6}$). This is not as clear for the cadaver dependence ($H_{4,140} = 9.125$, $p = 5.805 \cdot 10^{-2}$). All obtained geometrical and additional data of each specimen, separated by tissue type, are included in the S3 File.

## Discussion

### Procedure evaluation

Indeed, as other research groups have pointed out, there is a large dependence of material properties on the harvesting location of biomechanical materials [26–30]. This also applies to the local test direction of the specimens, which for example for cancellous bone mostly results from the main trabecular direction [31, 32], for cortical bone from the orientation of the osteons and fibers [28, 33–35] and for soft tissue mainly from collagen fiber orientation [36, 37]. This makes it essential to standardize the harvesting and to create a reasonable procedure for documentation—these were the achieved main objectives of this study. Although there were already good approaches for plainer problems, such as for long bones, i.e. femora

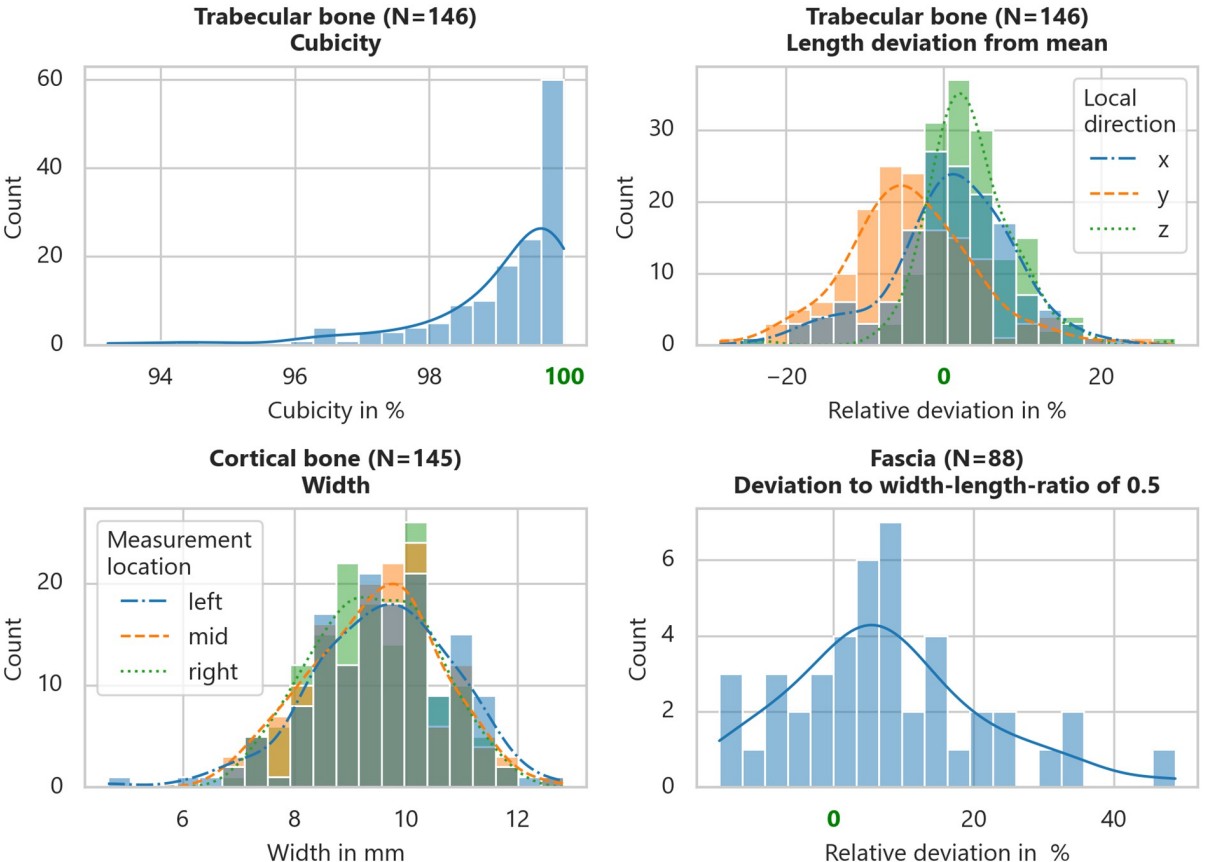

**Fig 8. Exemplary results of the specimens' geometrical quality control.** Results of different specimen types (*upper left*: trabecular bone cubicity distribution; *upper right*: trabecular bone length deviation of local directions from mean length; *lower left*: cortical bone width distribution by measurement location along intended span; *lower right*: fascia specimen width to test length ratio deviation of 0.5).

[28], this study provides a basis to also investigate complex structures such as the pelvis in a standardized way.

Reviews of biomechanical material parameters have been carried out in recent years. For example, published osseous material parameters were compared [38–41]. However, only Helgason et al. [41] and Öhman-Mägi et al. [39] were able to partially consider the preparation and condition of the specimens as quality criteria. In the studies examined, these were often not specified in detail or varied markedly. In addition to natural property variations, this could be another reason for the high variation of the presented material parameters.

The gold standard for testing would certainly be direct testing, immediately after specimen harvesting from the fresh cadaver preparation [23]. However, this is accompanied by the issue of practicability due to the unmanageable time requirement and the necessary high number of personnel. Nevertheless, this issue was solved sufficiently and reproducible by the established procedure for preparation and storage.

The intended minimization of physical, biological and chemical external influences on the specimens could also be achieved by strict execution of the developed procedure defined within the standard operating procedure. For example, due to harvesting, a thin band saw on frozen bone specimens and not as usual non-cooled hollow drilling at room temperature [30, 42] was used. This leads to the avoidance of burning, stagnating and unnecessary loss of

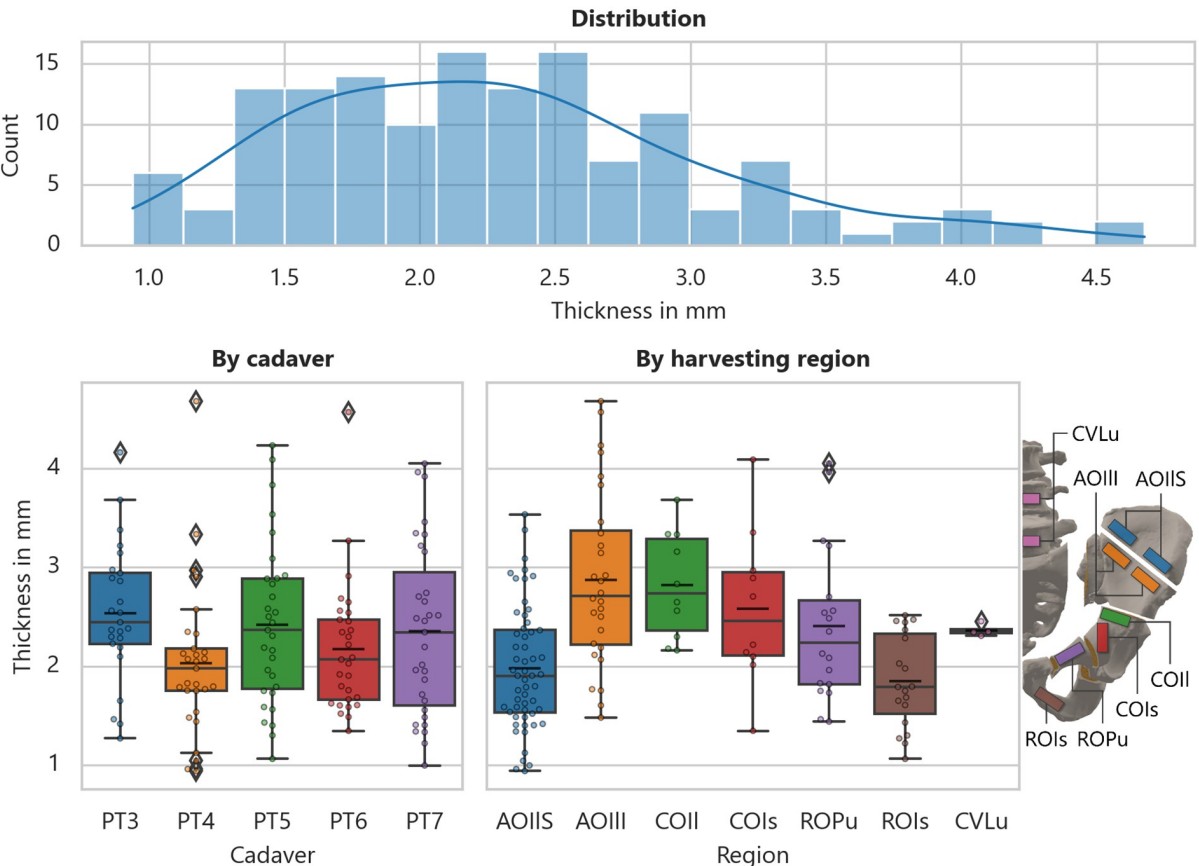

**Fig 9. Cortical bone thickness.** Observations on the mean thickness of the cortical bone specimens (*upper*: mean thickness distribution; *lower left*: mean thickness by cadaver; *lower right*: mean thickness by harvesting location with schematic locations).

material on the edges. Any post-processing of the specimens was also omitted. For example, mechanical treatment, like milling and grinding [28], of the cortical specimen surfaces would increase the shape quality but a priori severely degrade the surface integrity and thus contradict the principle of influence minimization. In a similar way, washing out the fat of the trabecular bone strongly influences material behavior by reducing damping characteristics, which is neither comprehensive nor reproducibly possible. Although in many areas of specimen preparation, it is indicated to use dimethyl sulfoxide solution (DMSO) [43] or other antifreeze agents. This was omitted due to the flotation of fascial and ligamentous tissues in order to preserve the mechanical and structural integrity of the cadaver [16].

Following the presented novel method, biomechanically relevant tissue specimens could be taken, which should allow after their testing a good material property mapping of the lumbopelvic system. The harvested specimens were controlled in accordance with the previously defined quality characteristics, especially regarding their geometry. The deviations determined here appeared to be minor considering the variability of biological materials. The standardized acquisition of relevant data and evaluation of process execution provide a seamless transition to material testing and its analysis. For this purpose, combined assessment codes were introduced, which allow a low-interpretation direct evaluation of the success or failure of the performed steps.

## Limitations

Precise alignment by surgical areas of interest might be slightly more beneficial in this regard. However, this was adapted in a few aspects due to reproducibility by finding visible and/or palpable anatomical landmarks, as well as in terms of manageability in the context of segment dissection.

Despite all the planning details, there is still a strong cadaver-specific sample dependency, due to previous injuries, diseases, as well as anatomical anomalies. In addition, possible damage to the tissue during specimen harvesting cannot always be completely avoided. Likewise, cadaver-specific differences can occur that prevent harvesting in more complex areas, even though the harvesting in the respective sections was indeed practicable in all test series and the harvesting areas were arranged in a way that this issue can be excluded as far as possible. This becomes apparent when the locally occurring natural cortical thickness of the human pelvis is examined, as shown in the results, especially in Fig 9.

Since, to our knowledge, there is as yet no standardized procedure for the comprehensive harvesting of human tissue for biomechanical testing, it is thus not possible for us to present evidence-based results due to the lack of comparative values. Therefore, at this stage, we can only propose an applicable foundation for standardization and thereby increase the reproducibility as well as the comparability of biomechanical tests.

## Conclusions

Following the general assumption that standardization of a process necessarily leads to a higher degree of reproducibility as well as enabling comparability in the first place, the presented approach could be a significant milestone in biomechanical material testing. The feasibility of the planned five-day specimen harvesting and testing cycle under the conditions of personnel requirements and reasonable practicability with regard to specimen storage could be demonstrated on five test and two pre-test sequences. In addition, the procedure structuring enables a better division of competencies among the processors in special teams based on the daily tasks. Thus, depending on the executing institution, separate teams can be deployed for preparation and testing tasks on corresponding days. A clear handover between these teams is thereby ensured by the developed protocols and assessment mechanisms. At the same time, however, it is also possible to conduct the entire procedure throughout with the same two persons to economize on personnel resources while at the same time maintaining the consistency of the results. Concurrently, the two-person-per-task approach minimizes the number of laboratory resources needed or used simultaneously.

In further research, additional investigation of the collagen distribution of involved structures, as well as the associated influxes of putrefaction on soft tissues, cortical bone, and trabecular bone would be beneficial. Furthermore, an adaptation of the three-dimensional sample topology planning via morphing could be beneficial to adapt it simply and rapidly to different pelvic geometries without having to manually adjust it each time.

All methods and procedures should be easily applicable to other biomechanical examination regions. Therefore, only the interesting regions, as well as the geometrical restriction of the testing methods, have to be defined. Anatomical landmarks have to be determined, whose connections define the sectioning planes required for segmentation. Consequently, the position of the specimen, local directions and designations can then be defined. The remaining part of the presented procedure can be kept unchanged thereafter. This is even partwise possible.

The methodology presented here has already allowed the acquisition of biomechanical material data of the human pelvis. As mentioned for the steps of the presented procedure, this

manuscript aims to keep the hurdles of applicability as low as possible. In the authors' opinion, this includes ensuring clarity, which is to be supported by a thematic separation. Therefore, an approach to standardized mechanical testing, evaluation and assessment of such specimens, will be independently considered in a forthcoming publication. This shall improve the reproducibility and comparability in this broad and interdisciplinary field, either in a combination of both procedures or individually.

## Supporting information

**S1 File. Standard operation procedure—Harvesting.** Including visual harvesting protocol for the lumbopelvic system, assessment code scheme as well as labels for storage boxes.
(PDF)

**S2 File. Models of auxiliaries.** Including 3D models and 3D-pdf overviews of preparation auxiliaries and storage boxes.
(ZIP)

**S3 File. Specimen data.** Tabular data of specimens including geometrical observations and additional information.
(XLSX)

## Acknowledgments

The authors would like to thank Prof. Dr.-Ing. Volker Slowik and Dr.-Ing. Thomas Klink from the Leipzig University of Applied Sciences for their continuous support.

## Author Contributions

**Conceptualization:** Sascha Kurz, Marc Gebhardt.

**Data curation:** Sascha Kurz, Marc Gebhardt.

**Formal analysis:** Sascha Kurz, Marc Gebhardt.

**Funding acquisition:** Sascha Kurz, Marc Gebhardt, Christoph-Eckhard Heyde.

**Investigation:** Sascha Kurz, Marc Gebhardt, Fanny Grundmann, Hanno Steinke.

**Methodology:** Sascha Kurz, Marc Gebhardt, Fanny Grundmann.

**Project administration:** Sascha Kurz, Marc Gebhardt.

**Resources:** Sascha Kurz, Marc Gebhardt, Christoph-Eckhard Heyde, Hanno Steinke.

**Software:** Marc Gebhardt.

**Supervision:** Hanno Steinke.

**Validation:** Sascha Kurz, Marc Gebhardt.

**Visualization:** Sascha Kurz, Marc Gebhardt.

**Writing – original draft:** Sascha Kurz, Marc Gebhardt.

**Writing – review & editing:** Sascha Kurz, Marc Gebhardt, Christoph-Eckhard Heyde, Hanno Steinke.

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
