## [Editor Report · Decision Letter 0]

14 Apr 2023

PONE-D-22-30924Approach to Standardized Material Characterization of the Human Lumbopelvic System - Specification, Preparation and StoragePLOS ONE

Dear Dr. Sascha Kurz,

Thank you for submitting your manuscript to PLOS ONE. After careful consideration, we feel that it has merit but does not fully meet PLOS ONE’s publication criteria as it currently stands. Therefore, we invite you to submit a revised version of the manuscript that addresses the points raised during the review process. 1. Lacks Bioethics Committee approval numbers.

2. Methodology -needs to be described in detail.

3. Results - structure the presentation of results, with subsections/subsections.

4. Discussion -needs to be developed.

Kind regards,

Malgorzata Wojcik, Ph.D

Academic Editor

PLOS ONE

3. "Thank you for stating the following financial disclosure:

“We acknowledge co-support from Federal Ministry for Economic Affairs and Energy (grant numbers MG: ZIM 16KN051655, SK: ZIM 16KN051656), the Saxonian State Ministry for Higher Education, Research and the Arts (stipend reference MG: 31004 70 809) and support from Open Access Publishing Fund of Leipzig University. The funders had no role in study design, data collection and analysis, decision to publish, or preparation of the manuscript.

Federal Ministry for Economic Affairs and Energy-ZIM: https://www.zim.de/ZIM/Navigation/DE/Home/home.html

Saxonian State Ministry for Higher Education, Research and the Arts - HTWK Leipzig - Förderlinie N-Promotion 2020: https://gradz.htwk-leipzig.de/

Open Access Publishing Fund of Leipzig University: https://www.ub.uni-leipzig.de/open-science/oa-allgemein/”

“We acknowledge co-support from Federal Ministry for Economic Affairs and Energy (grant numbers ZIM 16KN051655, ZIM 16KN051656), the Saxonian State Ministry for Higher Education, Research and the Arts (stipend reference 31004 70 809) and support from Open Access Publishing Fund of Leipzig University. The sponsors played no role in the study design, data collection, data analysis, data interpretation, or writing of the manuscript. Special thanks to Prof. Dr. Ing. Volker Slowik and Dr. Ing. Thomas Klink from the Leipzig University of Applied Sciences for their continuous support.”

“We acknowledge co-support from Federal Ministry for Economic Affairs and Energy (grant numbers MG: ZIM 16KN051655, SK: ZIM 16KN051656), the Saxonian State Ministry for Higher Education, Research and the Arts (stipend reference MG: 31004 70 809) and support from Open Access Publishing Fund of Leipzig University. The funders had no role in study design, data collection and analysis, decision to publish, or preparation of the manuscript.

Federal Ministry for Economic Affairs and Energy-ZIM: https://www.zim.de/ZIM/Navigation/DE/Home/home.html

Saxonian State Ministry for Higher Education, Research and the Arts - HTWK Leipzig - Förderlinie N-Promotion 2020: https://gradz.htwk-leipzig.de/

Open Access Publishing Fund of Leipzig University: https://www.ub.uni-leipzig.de/open-science/oa-allgemein/”

Additional Editor Comments:

Dear Dr Sascha Kurz,

1. Lacks Bioethics Committee approval numbers.

2. Methodology -needs to be described in detail.

3. Results - structure the presentation of results, with subsections/subsections.

4. Discussion -needs to be developed.
---

## [Author Response · Author response to Decision Letter 0]

3 May 2023

1. The changes have been made as requested and we hope the manuscript now meets your requirements.

2. The funding information was reviewed again and added to the cover letter as requested (page 2). Please let us know if we missed discrepancies.

3. The financial disclosure was reviewed again and added to the cover letter as requested (page 2).

4. All funding information was removed from the manuscript as requested and added to the cover letter (page 2). The acknowledgment section was therefore updated in the manuscript.

Additional editor comments:

1. There are no bioethics committee approval numbers, since the Institute of Anatomy of Leipzig University has a general institutional approval for the use of postmortem tissues of human body donors as declared in the manuscripts at the beginning of the methods section under preparation acquisition (lines 82-89):

“All tissues originated from the Institute of Anatomy of Leipzig University. While alive, all body donors gave their informed and written consent to the donation of their post-mortem tissues for education and research purposes. Being part of the body donor program regulated by the Saxonian Death and Funeral Act of 1994 (3rd section, paragraph 18, item 8), institutional approval for the use of the post-mortem tissues was obtained from the Institute of Anatomy, Leipzig University. The authors declare that all experiments were performed according to the ethical principles of the Declaration of Helsinki.”

Accordingly, this has already been formulated in a variety of publications, for example:

Hammer, Niels; Glätzner, Juliane; Feja, Christine; Kühne, Christian; Meixensberger, Jürgen; Planitzer, Uwe et al. (2015): Human vagus nerve branching in the cervical region. In: PloS one 10 (2), e0118006. DOI: 10.1371/journal.pone.0118006.

The separate section “Ethics Decleration” has been removed according to the E-Mail from May 3, 2023.

2. More details were added in lines 126-127, 151-172, 185-187, 195-196, 202-209, 215-216, 250-260 and 269-274.

3. Changed as requested in lines 288 and 313.

4. Developed and changed as requested in lines 348-357 and 380-391.

Reviewer's comments:

No additional comments were received.

PACE:

Uploaded, converted and replaced as requested.

Additional journal requirements:

1. Changed as requested in lines 128, 168, 180, 224, 241, 264, 301, 328 and 341.

2. Deleted as requested.

---

## [Decision Letter · Decision Letter 1]

17 Jul 2023

PONE-D-22-30924R1Approach to standardized material characterization of the human lumbopelvic system - specification, preparation and storagePLOS ONE

Dear Dr. Kurz,

Thank you for submitting your manuscript to PLOS ONE. After careful consideration, we feel that it has merit but does not fully meet PLOS ONE’s publication criteria as it currently stands. Therefore, we invite you to submit a revised version of the manuscript that addresses the points raised during the review process.

There is an additional request that the authors edit to add the specific anatomical landmarks for their cuts.

We look forward to receiving your revised manuscript.

Kind regards,

JJ Cray Jr., Ph.D.

Academic Editor

PLOS ONE

Journal Requirements:

Reviewers' comments:

Reviewer's Responses to Questions

**Comments to the Author**

1. If the authors have adequately addressed your comments raised in a previous round of review and you feel that this manuscript is now acceptable for publication, you may indicate that here to bypass the “Comments to the Author” section, enter your conflict of interest statement in the “Confidential to Editor” section, and submit your "Accept" recommendation.

Reviewer #1: All comments have been addressed

Reviewer #2: All comments have been addressed

2. Is the manuscript technically sound, and do the data support the conclusions?

Reviewer #1: (No Response)

Reviewer #2: Yes

3. Has the statistical analysis been performed appropriately and rigorously? 

Reviewer #1: (No Response)

Reviewer #2: I Don't Know

4. Have the authors made all data underlying the findings in their manuscript fully available?

Reviewer #1: (No Response)

Reviewer #2: Yes

5. Is the manuscript presented in an intelligible fashion and written in standard English?

Reviewer #1: (No Response)

Reviewer #2: Yes

6. Review Comments to the Author

Reviewer #1: The reviewer has over 18 years of research experience in biomechanical experiments using cadavers. I read the authors' paper with great interest. It is mentioned that the paper provides a detailed description of the aspects that all researchers in the current field are concerned about. However, these aspects are already well known to other researchers in the related field and are widely practiced and acknowledged. Hence, it appears challenging to highly evaluate the academic value of these papers.

Reviewer #2: The paper looks really good. The previous revisions have been completed. I am not a statistician so I did not comment on the stats however, they looked good to me. Only additional revision I have is to add specific anatomical landmarks for the boney pelvis cuts. Some of the specific landmarks were listed on page 16, lines 184-189, but it would be nice to know where all of the cuts were made. Especially since it was mentioned on page 16, line 177 that a "plane was drawn between each two landmarks". Finally I though figure 9 was awesome!

7. PLOS authors have the option to publish the peer review history of their article (what does this mean?). If published, this will include your full peer review and any attached files.

Reviewer #1: No

Reviewer #2: No

---

## [Author Response · Author response to Decision Letter 1]

18 Jul 2023

Response to Reviewers - Point-by-point

The authors would like to thank for the important and beneficial comments, all of them were considered. The manuscript has been revised accordingly and our responses can be found below.

Reviewer #1: To the authors

The reviewer has over 18 years of research experience in biomechanical experiments using cadavers. I read the authors' paper with great interest. It is mentioned that the paper provides a detailed description of the aspects that all researchers in the current field are concerned about. However, these aspects are already well known to other researchers in the related field and are widely practiced and acknowledged. Hence, it appears challenging to highly evaluate the academic value of these papers.

Answer to Reviewer #1:

Thank you for your honest and direct assessment. We are aware that many scientists are familiar with the difficulties of working with cadavers and try to solve them to the best of their knowledge and belief. However, the respective ways are unfortunately often very different and make good work sometimes difficult to compare. Our effort was therefore to show a way as simple as possible to address these problems while still achieving comparable results. So that many can contribute with their research to a comparable data set.

Reviewer #2: To the authors

The paper looks really good. The previous revisions have been completed. I am not a statistician so I did not comment on the stats however, they looked good to me. Only additional revision I have is to add specific anatomical landmarks for the boney pelvis cuts. Some of the specific landmarks were listed on page 16, lines 184-189, but it would be nice to know where all of the cuts were made. Especially since it was mentioned on page 16, line 177 that a "plane was drawn between each two landmarks". Finally I though figure 9 was awesome!

Answer to Reviewer #2:

Thank you very much for your constructive comments and encouraging words, we really appreciate it. For a better description of the sectioning planes we adjusted lines 176, 181, 184 and 186-188. Also we improved Fig 3 and added Table 1 for a clearer and more detailed description of the sectioning planes without overloading the text.

---

## [Editor Report · Decision Letter 2]

20 Jul 2023

Approach to standardized material characterization of the human lumbopelvic system - specification, preparation and storage

PONE-D-22-30924R2

Dear Dr. Kurz,

We’re pleased to inform you that your manuscript has been judged scientifically suitable for publication and will be formally accepted for publication once it meets all outstanding technical requirements.

Kind regards,

JJ Cray Jr., Ph.D.

Academic Editor

PLOS ONE
---

## [Editor Report · Acceptance letter]

25 Jul 2023

PONE-D-22-30924R2 

Approach to standardized material characterization of the human lumbopelvic system - specification, preparation and storage 

Dear Dr. Kurz:

I'm pleased to inform you that your manuscript has been deemed suitable for publication in PLOS ONE. Congratulations! Your manuscript is now with our production department. 

Kind regards, 

on behalf of

Dr. JJ Cray Jr. 

Academic Editor

PLOS ONE